



# Investigation of quartz electron spin resonance residual signals in the last glacial and early Holocene fluvial deposits from the Lower Rhine

**Marcus Richter and Sumiko Tsukamoto**

Department of Geochronology, Leibniz Institute for Applied Geophysics (LIAG), Stilleweg 2, 30655 Hanover, Germany

**Correspondence:** Marcus Richter (marcus.richter@leibniz-liag.de)

**Abstract.** In this study, we examined the residual doses of the quartz electron spin resonance (ESR) signals from eight young fluvial sediments with known luminescence ages from the Lower Rhine terraces. The single aliquot regenerative (SAR) protocol was applied to obtain the residual doses for both the aluminium (Al) and titanium (Ti) impurity centres. We show that all of the fluvial samples carry a significant amount of residual dose with a mean value of $1270 \pm 120$ Gy for the Al centre (including the unbleachable signal component), $591 \pm 53$ Gy for the lithium-compensated Ti centre (Ti-Li), $170 \pm 21$ Gy for the hydrogen-compensated Ti centre (Ti-H) and $453 \pm 42$ Gy for the signal that originated from both the Ti-Li and Ti-H centres (termed Ti-mix). To test the accuracy of the ESR SAR protocol, a dose recovery test was conducted and this confirmed the validity of the Ti-Li and Ti-mix signal results. The Al centre shows a dose recovery ratio of $1.74 \pm 0.16$, whereas the Ti-H signal shows a ratio of $0.56 \pm 0.17$, suggesting that the rate of signal production per unit dose changed for these signals after the thermal annealing. Nevertheless, all fluvial sediments investigated in this study carry a significant residual dose. Our result suggests that more direct comparisons between luminescence and ESR equivalent doses should be carried out, and, if necessary, the subtraction of residual dose obtained from the difference is essential to obtain reliable ESR ages.

## 1 Introduction

When sedimentary quartz was first investigated for electron spin resonance (ESR) dating 35 years ago by Yokoyama et al. (1985), a bleaching test was performed and an op-

tically unbleachable residual signal for the Al centre was detected. Moreover "zero age" samples were investigated, residual signals were detected and subsequently subtracted from the natural signal intensity to calculate the equivalent dose ($D_e$). This procedure led to ESR ages which were in good agreement with expected ages. Over the years, several bleaching experiments on quartz ESR signals were conducted and varying proportions of bleachable and unbleachable signal intensities for the Al centre were reported (e.g. Toyoda et al., 2000; Voinchet et al., 2003; Rink et al., 2007; Tsukamoto et al., 2018; Beerten et al., 2020). The Ti centre instead showed a better but varying optical bleachability depending on the monovalent charge compensator: the Ti-Na centre and the Ti-H centre were fully bleached within 24 h of artificial optical bleaching using a halogen lamp, whereas the Ti-Li centre was bleached within 72 to 168 h (Toyoda et al., 2000). Investigations of different samples revealed a significant variability in bleaching kinetics for both the Ti-Li and the Ti-H signals (e.g. Tissoux et al., 2007; Duval et al., 2017). The Ti centre is believed to be fully bleachable by sunlight exposure (e.g. Toyoda et al., 2000; Tissoux et al., 2007). So far, very few studies have reported residual doses of the quartz ESR signals from young or modern analogue samples, which could be directly comparable with the quartz optically stimulated luminescence (OSL) $D_e$ values. Beerten et al. (2006) found a total of 55 Gy (Ti-Li) for the youngest sample in a aeolian sedimentary profile and see this as a strong indicator of an unbleachable or unbleached residual dose. Tsukamoto et al. (2017) used modern aeolian quartz samples, whose OSL signal is well bleached, to investigate the bleachability of the ESR signals. They found large and varying residual doses for both the Al and Ti centres:

from 130 to larger than 1700 Gy for the Al centre (including the unbleachable signal component) and from 60 to 460 Gy for the Ti centre. They thus emphasised the importance of subtracting the residual dose, not only for the Al centre but also for the Ti centre. Timar-Gabor et al. (2020) measured the residual dose of aeolian samples from Australia and Ukraine, which have reported OSL $D_e$ values. For all samples, the ESR residual doses were found to be significantly larger than the OSL $D_e$, with the Al centre (also with unbleachable signal component) ranging from 480 to 700 Gy and the Ti centre ranging 100 to 580 Gy, highlighting the necessity of performing a residual dose subtraction. Although studies were done on dating fluvial sediments using ESR (e.g. Yokoyama et al., 1985; Laurent et al., 1998; Bahain et al., 2007; Tissoux et al., 2007, 2008; Duval et al., 2015, 2020; Bartz et al., 2018; Voinchet et al., 2019; del Val et al., 2019), the potential effects of the residual signals before deposition in both the Al centre and Ti centre have not been well investigated. Voinchet et al. (2015) introduced a bleaching index for various fluvial and aeolian sediment samples, and a very small residual dose of 4–28 Gy, after subtracting the unbleachable signal of the Al centre, has been reported. Toyoda et al. (2000) conducted a comparison of the signal bleachability derived from multiple signals. Based on the result, they reported quartz ESR intensities from multiple centres with different bleachability. An agreement of the ages can confirm that the signals were well bleached before deposition. Since then, this so-called "multiple centres" approach has been applied in several studies (e.g. Duval et al., 2015, 2017; Bartz et al., 2018, 2020). A similar comparison was also conducted between the quartz ESR ages and feldspar post-infrared stimulated luminescence (IRSL) or quartz thermally transferred (TT-) OSL ages (Bartz et al., 2019, 2020).

Another important issue which affects the accuracy of ESR dating is the ability of the measurement protocol to recover a known dose (Murray and Wintle, 2003). Previously, ESR dose recovery tests have been conducted by Beerten et al. (2008) on quartz derived from dune sands and Asagoe et al. (2011), who used quartz from tephra samples. Unfortunately, both studies use an intensive thermal treatment (annealing) of the sample to erase the natural signal before artificial irradiation, which reduces the significance of the test. Tsukamoto et al. (2017) applied a SAR-SARA (single aliquot regeneration and added dose; Mejdahl and Bøtter-Jensen, 1994) procedure for unheated modern sediments and used a slope between the added dose on top of the natural dose and the measured dose as a surrogate for the dose recovery ratio (Kars et al., 2014). A similar method was also adopted by Toyoda et al. (2009) and Fang and Grün (2020), who plotted the relationship between the added dose on natural aliquots and the increase in the apparent dose.

This study aims to investigate the size of the residual doses for the quartz Al and Ti centres in fluvial sediments using eight samples with known OSL ages (Lauer et al., 2011). In this study, we define the residual dose as the ESR $D_e$ values minus the OSL $D_e$ of the same sample, and this includes both bleachable and unbleachable components of the Al centre. These young sediments are investigated using the ESR SAR protocol and its performance is monitored by conducting dose recovery tests.

## 2 Samples

Fluvial sediments from Lauer et al. (2011) are from five gravel pits on either side of the lower terraces of the Rhine (Frechen, 1992) covering a clearance of 90 km from Niederkassel to Rheinberg, North Rhine-Westphalia, were used in this study. All sediments originated from the younger lower terrace of the Rhine River. A brief description of the samples is given in Table 1 and a detailed description of the sedimentary environment is given in Lauer et al. (2011). Previous work from Lauer et al. (2011) provides OSL $D_e$ using SAR protocol in the range of several tens of Gray (see Table 2). They used IR-stimulated and yellow-stimulated luminescence signals of potassium-rich feldspar as well as OSL of quartz to date a total of 11 samples. Mean quartz OSL $D_e$ values range from $14.8 \pm 0.3$ to $33.3 \pm 1.4$ Gy with dose rates in the range of $1.48 \pm 0.15$ to $2.41 \pm 0.18$ Gy kyr$^{-1}$. The mean OSL ages range from $8.6 \pm 0.5$ to $16.0 \pm 1.3$ ka (see Table 3). Thus, the sediments are Holocene or late Pleistocene age rendering them to be treated as young samples for ESR residual measurements. All samples show the Al and Ti centres, but three samples (ALH-I, ALH-II and MHT-I) showed a broad and strong, overlapping signal, presumably arising from paramagnetic Mn$^{2+}$ and Fe$^{3+}$ impurities. Eventually, eight samples of a grain size ranging from 100 to 250 µm were used to conduct ESR measurements. These are exactly the same samples that Lauer et al. (2011) used. No additional preparation steps were taken.

## 3 ESR measurements

A Bruker ELEXSYS E500 X-band ESR spectrometer with a variable temperature controller was used to run all measurements. The temperature inside the ER4119HS cavity was kept at 100 K through the evaporation of liquid nitrogen. The measurement settings for the detection of the Al centre [AlO$_4$]$^0$ were $335 \pm 15$ mT scanned magnetic field, modulation amplitude 0.1 mT, modulation frequency 100 kHz, 40 ms conversion time and 122.9 s sweep time and three to five scans. For the Ti centre [TiO$_4$ / M$^+$]$^0$, the settings were $350 \pm 5$ mT scanned magnetic field, modulation amplification 0.1 mT, modulation frequency 100 kHz, 30 ms conversion time and 61.4 s sweep time and 5–10 scans of the spectra. For all measurements, the microwave power was kept at 10 mW and the sample size was 60 mg. The light exposure of the quartz grains within the ESR quartz-glass sample tubes was kept at a minimum during the heating, artificial irradiation and ESR measurements. Furthermore, sample tubes

**Table 1.** Sample description after Lauer et al. (2011).

| Sample ID | Description |
|---|---|
| RB-I | cross-bedded sand with small amounts of Laacher See tephra |
| RB-II | horizontally laminated, well-sorted fluvial sand |
| MHT-II | horizontally laminated sand |
| MHT-III | horizontally laminated sand |
| LB-I | horizontally layered sand |
| NK-I | cross-bedded sand layers |
| NK-II | overbank deposits |
| ALH-III | fluvial sand, more gravel-rich with clay clasts |

**Table 2.** Mean ESR equivalent doses ($D_e$) and residual doses of the four signals compared with the mean OSL $D_e$.

| Sample | Equivalent dose | | | | Residual dose | | | | |
|---|---|---|---|---|---|---|---|---|---|
| ID | Al* | Ti-Li | Ti-mix | Ti-H | Al[a] | Ti-Li | Ti-mix | Ti-H | OSL[b] |
| | (Gy) | (Gy) | (Gy) | (Gy) | (Gy) | (Gy) | (Gy) | (Gy) | (Gy) |
| RB-I | $1314 \pm 16$ | $661 \pm 5$ | $496 \pm 36$ | $217 \pm 39$ | $1296 \pm 17$ | $643 \pm 5$ | $478 \pm 36$ | $199 \pm 39$ | $18.4 \pm 0.4$ |
| RB-II | $1235 \pm 8$ | $627 \pm 10$ | $540 \pm 10$ | $246 \pm 27$ | $1220 \pm 8$ | $612 \pm 11$ | $526 \pm 11$ | $231 \pm 27$ | $14.8 \pm 0.3$ |
| MHT-II | $1266 \pm 12$ | $659 \pm 2$ | $553 \pm 50$ | $292 \pm 33$ | $1237 \pm 13$ | $630 \pm 3$ | $524 \pm 51$ | $264 \pm 34$ | $28.8 \pm 1.3$ |
| MHT-III | $1543 \pm 36$ | $691 \pm 28$ | $468 \pm 29$ | $146 \pm 42$ | $1516 \pm 37$ | $664 \pm 29$ | $441 \pm 30$ | $119 \pm 43$ | $27.0 \pm 0.8$ |
| LB-I | $1963 \pm 82$ | $893 \pm 13$ | $677 \pm 127$ | $202 \pm 33$ | $1930 \pm 83$ | $859 \pm 15$ | $643 \pm 129$ | $169 \pm 35$ | $33.3 \pm 1.4$ |
| NK-I | $1086 \pm 6$ | $413 \pm 19$ | $448 \pm 5$ | $189 \pm 27$ | $1057 \pm 8$ | $384 \pm 21$ | $419 \pm 7$ | $160 \pm 29$ | $28.9 \pm 2.0$ |
| NK-II | $961 \pm 18$ | $517 \pm 31$ | $292 \pm 73$ | $150 \pm 31$ | $931 \pm 19$ | $487 \pm 32$ | $262 \pm 74$ | $120 \pm 32$ | $30.0 \pm 1.0$ |
| ALH-III | $1009 \pm 13$ | $467 \pm 19$ | $353 \pm 31$ | $115 \pm 33$ | $989 \pm 14$ | $447 \pm 20$ | $333 \pm 32$ | $95 \pm 35$ | $20.1 \pm 1.2$ |

[a] Including unbleachable signal component. [b] Lauer et al. (2011).

were stored in opaque black plastic bags between measurements. During the measurements, meticulous care was taken to ensure that the sample quantity and sample tube positioning and measurement temperature always remained the same for all measurements. The quality factor ($Q$) of the cavity was always greater than 8000 during the runs. All the samples were rotated three times in the cavity to calculate the mean signal intensity and to take into account the angular dependence of the signal.

As suggested by Toyoda and Falguères (2003), the intensity of the Al centre was taken from the first ($g = 2.0185$) to the last peak ($g = 1.9928$), as depicted in Fig. 1a. The overlapping peroxy signal intensity was subtracted eventually by using the ESR signal intensity after annealing (step 4; see Table 4). The intensity of the Ti centre signals was evaluated from peak-to-baseline or peak-to-peak amplitude following Tissoux et al. (2008), Duval and Guilarte (2015) and Duval et al. (2017) (Fig. 1a and b). The intensity of the Ti-Li centre was taken from the baseline to the peak at $g_3 = 1.913$, although this may be affected by Ti-H centre (cf. Tissoux et al., 2008). The intensity of the Ti-H centre was calculated from the $g_3 = 1.915$ peak to the baseline. Duval and Guilarte (2015) used the peak-to-peak intensity at around $g_2 = 1.931$ (see Fig. 1a and b) originating from both Ti-H and Ti-Li centres (referred to as Ti-mix in this study). These three different measurement options for the Ti centre are equivalent to op-

tions D, C and B of Duval and Guilarte (2015), respectively. An in-house-built X-ray irradiator, consisting of a Spellmann XRB401 source, was used for all laboratory irradiations. The X-ray parameters were fixed to 200 kV and 2 mA, and the dose rate was calibrated to $0.052 \pm 0.004$ Gy s$^{-1}$ (Tsukamoto et al., 2021). For heating and annealing of samples, an in-house-built device was used (Oppermann and Tsukamoto, 2015). The dose response curve (DRC) was fitted to a single saturated exponential function using Origin 2017 without any weighting to calculate $D_e$.

## 4   Performance tests and equivalent dose

### 4.1   Preheat plateau test

The ESR SAR protocol (see Table 4), which has been tested and satisfyingly applied in previous studies in regards to the Ti centre (Tsukamoto et al., 2015, 2017, 2018; Richter et al., 2020), was used for all measurements. Prior to $D_e$ measurements, a preheat plateau test was carried out to assure only stable signals are used. The sample with the lowest quartz OSL $D_e$ was chosen for this test (RB-II; $14.8 \pm 0.3$ Gy). Temperatures were set to 160, 180, 200 and 220 °C. Additionally, an aliquot without heating treatment was used, which is referred to as 20 °C (room temperature). Heating time was 4 min for preheating and 120 min for annealing at

**Table 3.** External dose rates, ESR ages derived from $D_e$, residual ages before burial and mean OSL ages for comparison.

| Sample ID | Ext. dose rate[a] Gy kyr$^{-1}$ | Age (from $D_e$) | | | | Residual age before burial | | | | OSL |
| --- | --- | --- | --- | --- | --- | --- | --- | --- | --- | --- |
| | | Al[b] (ka) | Ti-Li (ka) | Ti-mix (ka) | Ti-H (ka) | Al[a] (ka) | Ti-Li (ka) | Ti-mix (ka) | Ti-H (ka) | (ka) |
| RB-I | $2.15 \pm 0.11$ | $611 \pm 32$ | $308 \pm 16$ | $231 \pm 21$ | $101 \pm 19$ | $603 \pm 32$ | $299 \pm 15$ | $222 \pm 20$ | $92 \pm 19$ | $8.6 \pm 0.5$ |
| RB-II | $1.67 \pm 0.08$ | $739 \pm 36$ | $375 \pm 19$ | $324 \pm 17$ | $147 \pm 18$ | $731 \pm 35$ | $367 \pm 19$ | $315 \pm 16$ | $138 \pm 18$ | $8.9 \pm 0.5$ |
| MHT-II | $2.41 \pm 0.18$ | $525 \pm 40$ | $273 \pm 20$ | $230 \pm 27$ | $121 \pm 16$ | $513 \pm 39$ | $261 \pm 20$ | $218 \pm 27$ | $109 \pm 16$ | $12.0 \pm 1.0$ |
| MHT-III | $2.28 \pm 0.26$ | $677 \pm 79$ | $303 \pm 37$ | $205 \pm 27$ | $64 \pm 20$ | $665 \pm 77$ | $291 \pm 36$ | $193 \pm 26$ | $52 \pm 20$ | $11.8 \pm 1.4$ |
| LB-I | $2.08 \pm 0.15$ | $944 \pm 79$ | $429 \pm 32$ | $325 \pm 66$ | $97 \pm 17$ | $928 \pm 78$ | $413 \pm 31$ | $309 \pm 66$ | $81 \pm 18$ | $16.0 \pm 1.3$ |
| NK-I | $2.01 \pm 0.10$ | $540 \pm 27$ | $206 \pm 14$ | $223 \pm 11$ | $94 \pm 14$ | $526 \pm 26$ | $191 \pm 14$ | $209 \pm 11$ | $80 \pm 15$ | $14.4 \pm 1.2$ |
| NK-II | $2.11 \pm 0.12$ | $455 \pm 27$ | $245 \pm 20$ | $138 \pm 35$ | $71 \pm 15$ | $441 \pm 27$ | $231 \pm 20$ | $124 \pm 36$ | $57 \pm 15$ | $14.2 \pm 0.9$ |
| ALH-III | $1.48 \pm 0.15$ | $682 \pm 70$ | $315 \pm 34$ | $239 \pm 32$ | $78 \pm 24$ | $668 \pm 68$ | $302 \pm 33$ | $225 \pm 32$ | $64 \pm 24$ | $13.6 \pm 1.6$ |

[a] Lauer et al. (2011). [b] Including unbleachable signal component.

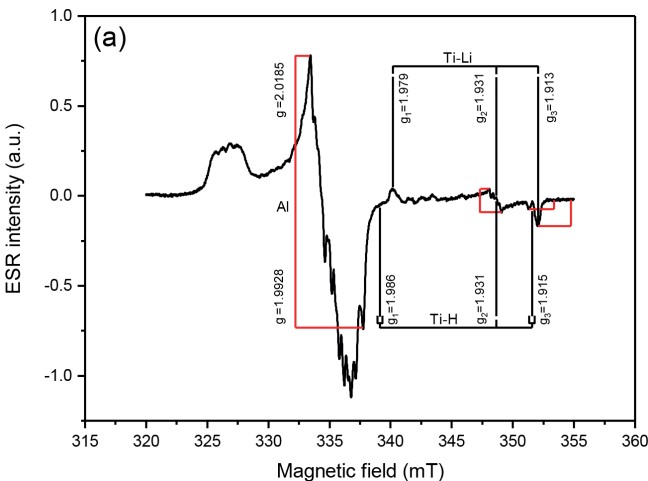

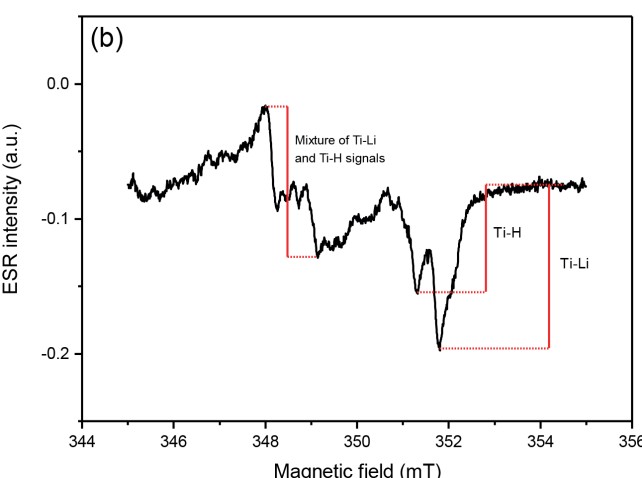

**Figure 1. (a)** The natural Al centre and Ti centres of sample RB-II and overview of the $g$ values; **(b)** close-up of titanium signals of sample RB-II after annealing and giving 500 Gy of artificial irradiation.

**Table 4.** ESR SAR protocol modified after Tsukamoto et al. (2015).

| Step | Treatment |
| --- | --- |
| 1 | Preheat ($T$ °C for 4 min)* |
| 2 | Natural ESR |
| 3 | Anneal (300 °C for 120 min) |
| 4 | ESR after annealing |
| 5 | Artificial irradiation |
| 6 | Preheat ($T$ °C for 4 min)* |
| 7 | Regenerated ESR |
| 8 | Repeat steps 5–7 |

* $T$ is the preheat temperature.

300 °C. In a previous study, Tsukamoto et al. (2015) compared 420 °C for 2 min and 300 °C for 120 min annealing time and found no significant difference in sensitivity change between both temperatures. Artificial irradiation dose steps used were 241, 963 and 2889 Gy to construct a dose response curve. The results are plotted in Fig. 2a. The $D_e$ value of the Al centre was initially decreased by the preheat at 160 °C but shows a steady increase in $D_e$ with increasing preheat temperature. At 220 °C, no $D_e$ calculation was possible, because all regenerated signal intensities were below the natural. The Ti-Li and Ti-mix signals show a similar pattern in $D_e$; there was a small decrease from room temperature to 160 °C, but all preheats yielded similar $D_e$ values, albeit a slight increasing trend with increasing temperature was observed. The Ti-H centre showed an opposite trend to Ti-Li and Ti-mix and showed a decrease in $D_e$ with higher temperatures > 180 °C. Eventually, the preheat temperature was set to 160 °C for all of the following measurements because Ti-Li, Ti-H and Ti-mix $D_e$ tend to form a plateau in the region of the 160–180 °C preheat temperature. An overview of the DRCs for 160 °C is shown in Fig. 2a, and each preheat temperature for each of the ESR centres can be found in Fig. S1 in the Supplement.

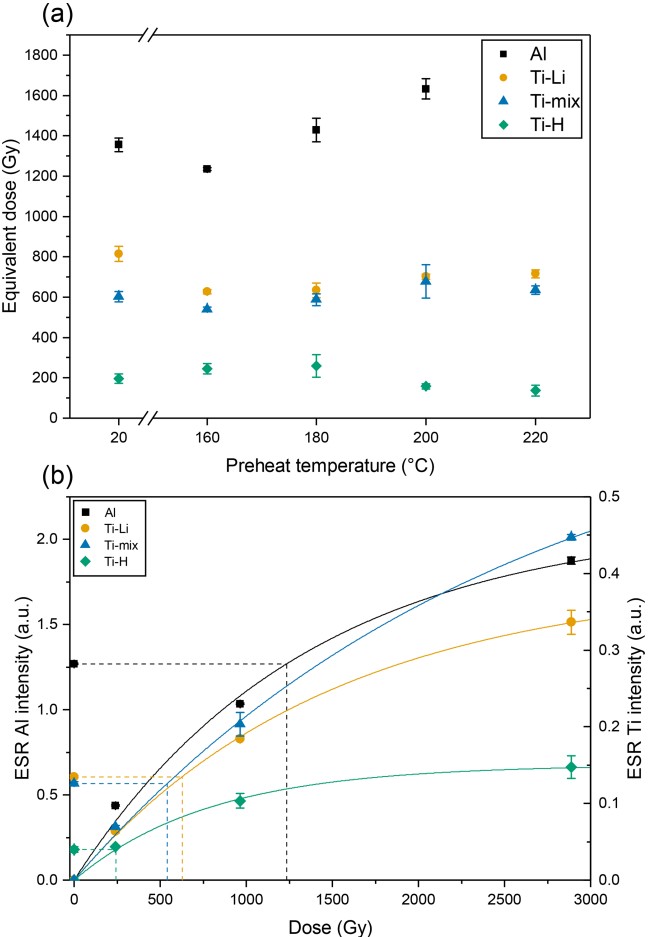

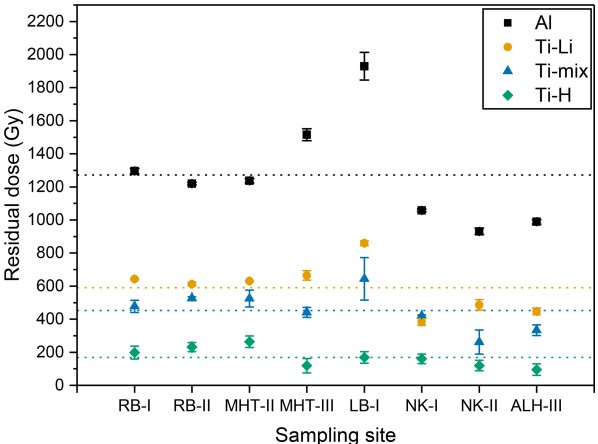

**Figure 3.** Residual doses of the four different ESR signals for all samples. Dotted lines indicate the mean dose for each signal.

mean of $453 \pm 42$ Gy, and Ti-H goes from 95 to 264 Gy with a mean of $170 \pm 21$ Gy. A detailed overview is given in Table 2. Residual doses of the four different ESR signals for all samples are plotted in Fig. 3. A detailed list of ages is given in Table 3. All the ESR ages significantly overestimate the OSL ages. The ages (calculated from the residual dose) are on average $634 \pm 54$ ka for Al centre (including the unbleachable signal component), $294 \pm 25$ ka for the Ti-Li, $227 \pm 22$ ka for the Ti-mix and $84 \pm 10$ ka for the Ti-H centre. These residual ages show how significant the effect of the residual dose may be in ESR dating of fluvial sediments.

### 4.3 Dose recovery test

A dose recovery test, using the SAR protocol, was performed for all four ESR signals by adding 963 Gy on top of the natural signal using three aliquots of sample RB-II and thus is considered to be a new "natural" signal. The test was used to check the accuracy of the measurement protocol because the thermal treatment included in the SAR protocol may change sensitivity of the ESR centres. The $D_e$ values of the aliquots (natural + 963 Gy) were measured by the SAR protocol, with three dose steps up to 3516 Gy. The dose recovery ratio was calculated by subtracting the natural $D_e$ from the recovered dose and the difference of the natural + 963 Gy and the natural $D_e$ was then divided by the added dose of 963 Gy. This experiment is a modified version of the single aliquot regenerative and added dose (SARA) by Tsukamoto et al. (2017) with a single added dose point. The dose recovery results (see Fig. 4) are satisfactory for the Ti-Li and Ti-mix signals with a ratio of $0.98 \pm 0.07$ and $1.00 \pm 0.15$, respectively, indicating that ESR SAR protocol works well for these signals. Our results resemble the results published by Tsukamoto et al. (2017). The dose recovery ratio for the Al signal is high with $1.75 \pm 0.18$, whereas the ratio of the Ti-H signal is low $(0.55 \pm 0.17)$. The significantly smaller Ti-H $D_e$ compared

**Figure 2. (a)** Preheat plateau test for sample RB-II. The dose response curve for Al centre for 220 °C did not fit, so the $D_e$ value was not obtained. **(b)** Dashed lines indicate the mean dose for each signal. **(b)** The DRCs for 160 °C preheat temperature for each one of the ESR centres. The $D_e$ values are marked.

### 4.2 Equivalent doses, residual doses and ESR ages

For each sample, one aliquot was used to conduct the $D_e$ measurements. Dose response curves were created using three regenerated dose steps with a total dose up to 2889 Gy for all samples except for samples NK-1, NK-2 and ALH-III, which were irradiated up to 3022 Gy. The $D_e$ values of the Al centre are in the range of 961 to 1960 Gy (including the unbleachable signal component). The $D_e$ values of the Ti-Li centre span from 413 to 893 Gy. The Ti-mix $D_e$ ranges from 292 to 677 Gy, and the Ti-H $D_e$ goes from 115 to 292 Gy. The mean OSL $D_e$ for each sample was subtracted from the ESR $D_e$ to calculate the residual dose. This led to a residual dose of Al centre in the range of 931 to 1930 Gy with a mean value ($\pm 1$ SE) of $1270 \pm 120$ Gy (including the unbleachable signal component). The Ti-Li centre residual dose goes from 384 to 859 Gy with a mean of $591 \pm 53$ Gy. The Ti-mix residual dose goes from 262 to 643 Gy with a

https://doi.org/10.5194/gchron-4-1-2022

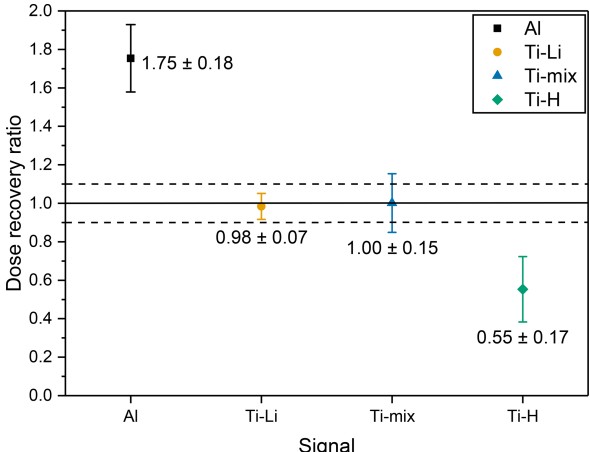

**Figure 4.** Dose recovery ratios. The dashed lines mark the 10 % deviation margin.

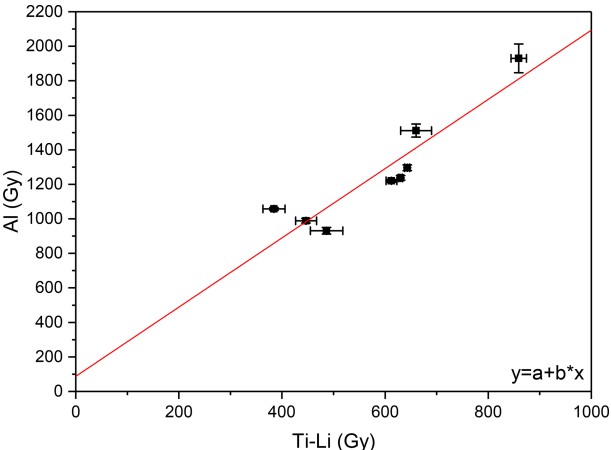

**Figure 5.** Comparison of ESR Al and Ti-Li residual doses with linear fitting.

to the Ti-Li $D_e$ is probably partly a result of this (underestimating). The result of our dose recovery test suggests that the applied SAR protocol is robust in the dose estimation for the Ti-Li and Ti-mix signals, whereas those from the Al and Ti-H centres could be over- and underestimated.

## 5 Discussion and conclusion

The results clearly show that the ESR $D_e$ for all samples are significantly larger than the OSL $D_e$ of Lauer et al. (2011), and therefore residual subtraction is highly recommended if a representative modern analogue sample is available. Furthermore, the observed residual doses follow the trend in the signal's bleaching behaviour as described by Toyoda et al. (2000): the Al centre shows the largest residual followed by Ti-Li and Ti-H with the lowest residuals. The size of the residual dose for the Ti-mix lies in between those of Ti-Li and Ti-H. However, it should be noted that the recovered dose in the dose recovery test overestimated the given dose for the Al centre and showed underestimation for the Ti-H centre, which may have influenced the observed residual dose. Although the Ti-H shows the smallest $D_e$, and hence is closest to the expected OSL $D_e$, it is unreliable because it failed to recover the known given dose.

Regarding the Al centre, we did not estimate the size of the bleachable and unbleachable components by a bleaching test. Instead, a measured residual dose from young samples, preferably obtained from the same set of sedimentary sequence, could be subtracted from the $D_e$ of older samples; this approach has an advantage over the very time-consuming bleaching experiment with the solar simulator for $\sim 1000$ h. Figure 5 shows a comparison of all residual doses for the Al and Ti-Li. Additionally, a linear fitting was performed yielding the $y$ intercept of $90 \pm 220$ Gy. This intercept indicates a rough estimate of the size of residual dose for the unbleachable Al centre, although it is much smaller than the values

reported by Tsukamoto et al. (2018) and Timar-Gabor et al. (2020) from aeolian sediments.

However, the result of the dose recovery test suggests that the thermal annealing step in the SAR protocol changed the signal production efficiency of the Al centre. We hypothesise that the annealing changed the ratio of bleachable and unbleachable components of the Al centre, which led to the failure of the dose recovery test. Timar-Gabor et al. (2020) demonstrated that the intensity of both the bleachable and unbleachable components of the Al centre can be increased by additive dose irradiation on natural aliquots. They explained that the Al centre has an unbleachable component, because the amount of Al in quartz is far more abundant compared to any other electron centres, which contributes to bleaching (and recombines with the Al-hole centre). However, a thermal annealing reset both populations, and following irradiation it may have only produced the bleachable Al centre. Although this hypothesis must be tested experimentally, supporting evidence of the hypothesis is available from a comparison of the natural and regenerative dose response curves of the Al centre from the Chinese Loess Plateau. Tsukamoto et al. (2018) showed that the regenerated dose response curve, which was constructed after an annealing, was only comparable to the natural one, when the unbleachable Al signal intensity was subtracted from the natural dose response curve, suggesting that the regenerative dose response curve was dominated by the bleachable Al centre.

The dose recovery test of the Ti centre indicates that Ti-Li centre does not suffer any sensitivity changes after the annealing, whereas the Ti-H centre underestimates the given dose significantly. Beerten and Stesmans (2006) reported strong deviations in Ti-Li and Ti-H SAR $D_e$ from the expected dose, although the total Ti centre provided a reliable result. They suggested different possible explanations including (1) charge transfer between Ti-Li and Ti-H centres during the artificial irradiation and (2) differences in production ef-

ficiency, but eventually they left the question open. Similar problems might have also affected the observed difference in the dose recovery ratios of the different Ti signals. More effort is needed to fully understand about the behaviour of different Ti signals.

Though available sedimentological information for the samples is limited, we compared the observed residual dose in different fluvial depositional environments affected. From Lauer et al. (2011), we identified three different depositional environments, which include (i) overbank deposits, (ii) deposits from braided river systems and (iii) deposits of a channel, i.e. meandering river system (Michael Kenzler, personal communication, 2021). The Rheinberg samples (RB-II and I) were taken from a point-bar setting and have been interpreted as channel deposits of a meandering river. The samples from Monheim–Hitdorf were deposited in a braided river system with channel and sheet flow deposits. At Libur, sample LB-I originated from a braided river system. The Niederkassel site sample (NK-I) was deposited in a braided river system, whereas the lower NK-II sample seems to have originated from an overbank deposit. The Aloysiushof/Dormagen sample (ALH-III) stems from the uppermost gravel-rich part of the profile and probably channel deposits.

From the observed residual doses, we do not see any pattern according to different depositional environments. Instead, all residual doses for our samples are relatively uniform, with a mean of $1270 \pm 120$ Gy for the Al centre (including the unbleachable signal component), $591 \pm 53$ Gy for the Ti-Li centre, $170 \pm 21$ Gy for Ti-H and $453 \pm 42$ Gy for Ti-mix.

In conclusion, we show that all of the investigated fluvial sediments were not fully bleached before burial, and after subtraction of OSL $D_e$ a significant amount of the residual dose was still carried by the samples. Even the Ti-H, which is supposed to be the most bleachable, is far from zero. This highlights the importance of further investigation into the dynamics of residual doses in both aeolian and fluvial environments.

**Data availability.** All data generated or analysed during this study are included in this published article.

**Supplement.** The supplement related to this article is available online at: https://doi.org/10.5194/gchron-4-1-2022-supplement.

**Author contributions.** MR and ST conceived the study; MR carried out the measurements with input from ST. MR wrote the paper with input from ST.

**Competing interests.** At least one of the (co-)authors is a member of the editorial board of *Geochronology*. The peer-review process was guided by an independent editor, and the authors also have no other competing interests to declare.

**Disclaimer.** Publisher's note: Copernicus Publications remains neutral with regard to jurisdictional claims in published maps and institutional affiliations.

**Acknowledgements.** We are grateful to Gwynlyn Buchanan for language corrections and Michael Kenzler for sedimentological interpretation. The constructive comments from three reviewers, Mathieu Duval and two anonymous reviewers, helped to improve the paper.

**Review statement.** This paper was edited by Georgina King and reviewed by Mathieu Duval and two anonymous referees.

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
