# Peer review of "Investigation of quartz ESR residual signals in the last glacial and early Holocene fluvial deposits from the Lower Rhine"

_Geochronology, 2021_

## Author Comment (AC1)

The study of Richter and Tsukamoto entitled "Investigation of quartz ESR residual signals in the last glacial and early Holocene fluvial deposits from the Lower Rhine" presents original results that are very important for the ESR dating community. The manuscript is well written. The main message is that Al-h as well a Ti-Li and Ti-H signals in quartz are not fully reset in the case of fluvial sediments. By analyzing Holocence samples that were securely dated by OSL for ESR dating the authors report a mean value of 1350 ± 120 Gy for the residual of Al centre, 610 ±60 Gy for the lithium-compensated Ti centre (Ti-Li), 170 ± 20 Gy for the hydrogen-compensated Ti centre (Ti-H), and 470 ±50 Gy for the signal originated from both the Ti-Li and Ti-H centres (termed Ti-mix), concluding that fluvial sediments carry a significant residual dose, and therefore the subtraction of residual dose using a modern analogue is highly recommended to obtain reliable ESR ages. I fully concur with the main message of the paper and I think it is worth being published.

We thank the reviewer for the very insightful comments to our manuscript.

However, besides this, the study presents yet another problem, namely the poor dose recovery results. The way the dose recovery experiment was conducted is problematic in my view and leads to circular reasoning. If the authors cannot prove that my observation is wrong, the interpretation provided now needs to be revised. The approach of the dose recovery test was to use the "natural" sample that holds this significant residual, irradiate it with 1000 Gy and as the authors explain "The dose recovery ratio was calculated by subtracting the natural De from the recovered dose and the difference of the natural + 1000 Gy and the natural De was then divided by the added dose of 1000 Gy." The problem is that by doing so, it is inherently assumed that the determined De is correct.

While this would not be a problem in a luminescence experiment with a residual of say 10 Gy compared to the given dose of 1000 Gy, here it is, because the magnitude of the residual, in this case the De is of the same order of magnitude as the given dose. The authors then state that "The dose recovery ratio for the Al signal is high with 1.74 ± 0.16, which indicates a sensitivity change due to thermal treatment during SAR protocol, therefore the reported residual doses may be overestimated." But to my understanding, first it was assumed accurate based on your experiment or am I missing something?

And you subtracted this value in the numerator of your ratio. The way this experiment stands defies in my view the idea of a dose recovery test as the given dose should be known and the equivalent dose is by definition an unknown. The only way in which this dose recovery could work is if the "sensitivity" is the same in the De (or residual dose here) measurement and during the dose recovery experiment. Which brings me to the point when I have to mention that I find the term "sensitivity change" to be extremely difficult to be conceptually integrated in ESR dating.

Anyway, conceptually the problem is that you draw a conclusion on the accuracy of the measurement protocol (namely that it is not very accurate) by feeding into your protocol a value measured by the same protocol that is assumed to be accurate.

Although we understand the criticism of the reviewer, implementing dose recovery has been something really missing in ESR dating to prove the robustness of measurement protocols.

The current version of dose recovery test is a simplified one of Tsukamoto et al. (2017), who conducted the single aliquot regenerative and added dose (SARA), after Mejdhal et al (1994) and Kars et al. (2014) using a single added dose point. Here, dose recovery ratio is the slope between SAR De and added dose. Assuming a same problem (if there is) is affecting the performance of the SAR protocol in both natural and natural + 1000Gy measurements, the slope should be equivalent to the dose recovery ratio. And since the full SARA measurement is very time consuming, the one point SARA as a dose recovery test is the best thing we can think of.

My second problem concerns figure 5 where the residual of Al-h is plotted as function of that of Ti-Li. While I fully agree with the logic of this approach and I consider it a very elegant way to extract the unbleachable part of the Al-h signal, even by eye one can see that that line does not fit the data. I plotted the data myself and obtained an intercept of 134 ± 208 Gy by simple linear fitting so I cannot understand how that regression was obtained. The result I have quoted above is of course very concerning, as I understand that the authors know from experience, by the simple examination of the signals measured on modern analogues from aeolian environments that Al-h is not fully bleachable. I fully support this view. I think that the fact that the intercept is close to zero and the scatter in your data simply reflects the problems associated with the measurement protocol, as mentioned in my comments above and below.

We admit that the linear trend line was drawn incorrectly. After reviewing the data, we agree with the trend line/intercept with the y-Axis of the reviewers comment. The figure was adjusted and the text was also adapted.

While one should not expect this problem to be solved in the framework of this particular study the observation "The De value of the Al centre was initially decreased by the preheat at 160 °C, but shows a steady increase in De with increasing preheat temperature. At 220 °C no De calculation was possible, because all regenerated signal intensities were below the natural" is extremely concerning. I suggest the authors show some of the measured signals and the constructed dose response curves in the supplementary file.

[Figure]

The figure above shows the DRCs for each preheat temperature (RT = room temperature = 20 °C) for each one of the ESR centres. The DRC's for the Al signal (d) show a systematic flattening in the DRC due to lower signal intensities achieved by the same dose applied with a higher preheat

temperature. In case of a preheat temperature of 220 °C the intersection from DRC and the natural signal intensity is > 3000 Gy. This figure has been added to the supplement (Figure A1).

At this point one has to ask himself why one needs to preheat in ESR dating? How come a single type of defect can have a part that is not thermally stable and one that is not, as mentioned in Toyoda and Ikeya, 1994 QG, and explained here as "a preheat plateau test was carried out to assure only stable signals are used"?

We think it is possible that after artificial irradiation (with a few order of magnitude higher than in nature) there can exist thermally unstable Al and Ti centres. Thus it is beneficial at least to conduct a mild preheat to check if it is not the case. Our previous studies (Tsukamoto et al., 2015; 2018) showed the need of preheating.

Especially in our study of the Chinese loess-palaeosol sequence (Tsukamoto et al. 2018, RM) when we compared artificial and natural DRC, that the signal intensity and $D_0$ of the laboratory ones are much larger than the natural ones. Both decrease with preheat and match better with the natural DRC, confirming that a preheat step is necessary after artificial irradiation.

And how does that relate to the values of E and s quoted in studies such as Richter et al., 2020, QI? ESR is selective, as such only one type of defect is targeted. Does this mean that we acknowledge that both localized and delocalized transitions are at play? If the localized route is the one that leads to an apparent instability, then one should expect the degree of instability to be dose dependent (see Benzid and Timar-Gabor 2020, AIP Advances 075114). If indeed there is a degree of instability that is dose dependent that might explain the observation I am referring to above.

As described in Tsukamoto et al., 2018, we preheated at the same temperature (160 °C, 4 min) for all three aliquots before isothermal measurement started, and the normalised the intensity to the first preheated signal. So we want to make sure that anything thermally very unstable must have gone by the initial preheat.

My last comment concerns the interpretation "The opposite tendency of the heat treatment for the Ti-Li and Ti-H centres in the preheat plateau test (Fig. 2) suggests that some charge transfer between the Ti-Li and Ti-H centres is likely." Personally, I am unable to grasp what model is proposed. Please be more specific. What is the transferred charge?

We do not have enough evidence for a detailed discussion, so we have decided to remove the section cited by the reviewer from the MS.

Technical comment: line 70-reference to table 3 should be made instead of table2.

This reference has been changed to Table 4 since we added another Table in the beginning of the MS.

---

## Author Comment (AC2)

This is an interesting study that provides new data around the behaviour of the ESR signals in quartz from young (Late Pleistocene to Holocene) sediment. In particular, the authors evaluate whether the ESR signals associated to the Al and Ti centres are actually fully reset during sediment transport and prior to deposition. This is done by comparing the ESR dose estimates with those independently obtained from OSL measurements. As a matter of fact, the ESR dose values are significantly (one or two orders of magnitude) higher (between 150 and 1350 Gy depending of the centre considered) than the OSL ones (15 to 35 Gy). This indicates that the samples have not been fully bleached during transport, resulting in a massive dose (and age) overestimation if the unbleached component of the signal is not taken into account. Given these results, the authors recommend the use of modern-analogue samples to evaluate whether the ESR signals of Pleistocene samples have been fully reset.

The paper is very well written, easy to read and some of the conclusions are supported by the data displayed in this work, although the general take-home message should be in my opinion somewhat moderated (and especially the last sentence of the abstract: 'The results of this study suggest that fluvial sediments carry a significant residual dose, and therefore the subtraction of residual dose using a modern analogue is highly recommended to obtain reliable ESR ages.'). The conclusions may be valid for this specific set of samples, but should not be universally extrapolated to all samples in any sedimentary context. The present study is based on a limited number of samples, which all come from the same sedimentary environment, and supposedly have the same bleaching and deposition history. It is therefore not surprising that they all provide consistent results. The sedimentological characteristics of the sediment are known to possibly impact the level of optical bleaching achieved during transport (e.g., Voinchet et al., 2015). Moreover, there are many works in which ESR dating (using Al, Ti-Li and/or Ti-H signals) provided consistent results with independent age control based on Luminescence, Cosmogenic, Palaeomagnetism or Ar-Ar methods (see an overview in Duval et al., 2020b), for chronologies ranging from the Early Pleistocene to the Late Pleistocene. These should be considered as empirical evidence indicating that the ESR signals have indeed been fully reset (i.e., zeroed for the Ti signals, and reset to the unbleachable level for the Al signal) in frequent occasions, and that the so-called 'residual dose' is not significantly impacting the results. My point here is not to minimize the importance and relevance of the present results, but just to put them in a broader perspective. Whether the ESR signal is fully reset may depend on many factors, including the transport conditions, and nature and composition of the raw sediment. The present study is based on samples from a single environment that is perhaps simply not favourable to ensure complete bleaching of the ESR signals. Finally, there are several studies for which the De values derived from either Al or Ti centres are <100-150 Gy (see an overview in Bartz et al., 2020), i.e., below the dose estimates obtained in the present work. This indicate that the magnitude of this so-called unbleached (residual) component is sample- and environment-dependent. This aspect would deserve to be mentioned in the present work.

The above mentioned papers (Bartz et al., 2020 and Voinchet et al., 2015) have been referred in the introduction.

Additionally, although the 'residual dose' may be due to an incomplete bleaching of the ESR signals prior to deposition, other sources of uncertainty should also be explored, as they may impact the dose evaluation. For example, there is no mention at all of the real uncertainty associated to the ESR measurements (see point #2 in General comments), as well as of the potential bias simply induced by the way the ESR intensities are measured (see point #1). If the authors could explore and discuss these questions in the manuscript, that would surely bring added value to the study.

We have defined the residual dose in this study at the end of the introduction to avoid any confusion.

**General comments**

1. Identification of the ESR signals and measurement of ESR intensities.

There is still little consensus within the community around the way Ti ESR signals are identified, measured and reported. First of all, the Ti-H signal is made of a series of doublet, one of them being centred at g = 1.915. This one overlaps with the Ti-Li peak @ g = 1.913. Consequently, the peak reported as 'Ti-Li' by the authors on Figure 1B and throughout the manuscript (called option D in Duval and Guilarte, 2015) is actually resulting from the mixed contribution from Ti-H and Ti-Li signals (e.g., Toyoda et al., 2000). Therefore, this should be reported as 'Ti signal' or 'mix Ti', but not as Ti-Li (unless the authors can demonstrate that the Ti-H component has no impact on the peak). The only way to measure the single contribution of the Ti-Li centre is to measure the peak intensity @ g= 1.979 (Option E in Duval and Guilarte, 2015; Absorption line 1 in Beerten et al., 2020). If option E gives the same result as D, then it would be fair to consider that the Ti-H component has no significant influence on the peak intensity measured @ g = 1.913.

The option D of Duval and Guilarte, 2015 is still dominated by the Ti-Li centre and the contribution of the Ti-H centre was considered negligible (Duval et al. 2017). The potential influence of the Ti-H centre had been already mentioned in the text.

The authors decide to use an unusual way of measuring the Ti signal (so-called option B in Duval and Guilarte, 2015; Absorption line 2 in Beerten et al., 2020). This one is not frequently employed for dating purpose (I only recall Beerten et al., 2020). Therefore, their results are not directly comparable with all previous studies that are cited in the MS, since they are mostly based on Option A (as per reported in Duval and Guilarte, 2015) (e.g., Timar-Gabor et al., 2020; Rink et al., 2007) or option D (Tissoux et al., 2007). This is why I would encourage the author to report the data from options A and E (at least in SM), so that : (i) the present work can be directly comparable with other studies, and (ii) we can evaluate whether the dose results significantly differ depending on how the signal is measured.

When we started the measurements we did not plan to compare the residual dose of the Ti centre with different charge compensators. The Al and Ti centres were measured separately, using a small sweep width for the Ti centre, and unfortunately Opt A and E were not recorded (see Figure 1b). Figure 1a was only recorded once to show a complete spectrum in the publication. The correlation of our measurement options of the Ti centre to those of Duval and Guilarte (2015) has been added in the text of section 3.

2. Measurement procedure and uncertainty associated to ESR intensities.

Unlike for tooth enamel, ESR measurements of quartz samples at low temperature are usually characterized by a non-negligible uncertainty (at least a few %) that may result from a combination of different factors, such as the stability of the ESR spectrometer, the heterogeneity of the quartz samples, the angular dependence of the signal, the position of the tubes in the cavity, etc. Although ESR measurements are usually performed under controlled conditions, measurement precision for the Al and Ti signals is typically of at least a few %. This has a non-negligible impact on the $D_E$ values, which is why repeated measurements should be performed in order to evaluate the repeatability of the $D_E$ estimates (e.g. Duval et al., 2017, 2020a). A variability of at least 10% may be frequently achieved (at least with the MAAD method) and it may sometimes be >30%, casting thus in this case reasonable doubts on the reliability of the ESR data obtained.

During the measurements, meticulous care was taken to ensure that the measurement temperature, sample quantity, sample tube positioning, spectrometer stability and sample rotation procedure always remained the same for all measurements.

In their measurement procedure, the authors take into account the angular dependence of the signal by rotating the tube 3 times in the cavity. So I understand that they derived a mean ESR intensity value and associated error, which was then used for DRC fitting and dose evaluation (correct? please clarify in the MS). So, I wonder:

Yes, this is the way we calculated the mean ESR intensity and associated error. This information has been added to the MS in section "ESR measurements".

- About the magnitude of the experimental error derived from the rotations in the cavity ? This error may significantly vary depending on the signal considered. It would be interesting if the authors could comment on this.

Relative standard deviation (1-sigma, %) of the three measurements for each signal at different irradiation steps is shown below.

| In % of mean intensity | Al centre | (Al after subtraction of peroxy) | Ti-Li | Ti-mix | Ti-H |
|---|---|---|---|---|---|
| Natural | 0.6 | 0.9 | 2.4 | 8.4 | 16.4 |
| Dose 1 | 0.9 | 1.3 | 2.8 | 2.6 | 4.2 |
| Dose 2 | 0.8 | 0.9 | 2.3 | 5.3 | 5.3 |
| Dose 3 | 1.0 | 1.1 | 3.0 | 4.2 | 9.4 |

- In which extent this experimental error may impact the De value? This should be discussed.

The cause is probably the turning of the sample within the cavity. In the case of Ti-H the size of the experimental error is largest for the natural sample probably due to the relatively small natural signal intensity. The fitting of the dose response curves went well for all samples, a $R^2 > 0.99$ was generally achieved. Therefore, we can assume that the majority of the experimental error arises from the turning of the sample within the cavity.

Moreover, there is no Dose Response Curves (DRC) displayed. Can the authors provide some examples in the MS for the different signals measured ?

This Figure shows the DRCs for sample RB-II for the four used ESR signals at 160 °C preheat. This figure has been added (is Figure 2B).

[Figure]

Apart from the angular dependence of the signal, there is no mention of any repeated ESR measurements, while it is known that the variability of the ESR intensities over successive days of measurements may be significant. And this variability may be even higher when using a HS cavity (compared with a ST cavity). This may be especially crucial for the Ti-H signal, whose ESR intensity is the weakest of all signals, resulting in significant experimental errors that may sometimes exceed 10%. In other words, a poor De repeatability could possibly partly explain the dose results obtained in this work, which is why this should be evaluated for the present set of samples.

Finally, only 1 single aliquot has been measured per sample, but again, we have no idea of how representative is this aliquot of the whole sample. It seems essential to measure a few aliquots per sample in order to evaluate the dispersion of the results. I imagine this would be a standard procedure in OSL dating, and I would recommend to do it here in order to evaluate in which extent the homogeneity of the sample may impact dos estimates.

The robustness of our measurement procedure had been tested by the dose recovery test and the result was already discussed. Regarding intra-aliquot scatter, one aliquot (60 mg of quartz) contains a few thousand grains of quartz. So any grain to grain variabilities in pre-burial bleaching must have been well averaged. Moreover, the purpose of this study is to investigate the range of residual doses from a set of fluvial samples, not to date individual samples. We do not think measuring more aliquots would add significant more values in this paper.

I realise that these extra measurements might be time consuming, but they can actually be carried out much faster with the SAR method than the MAAD with given the limited number of dose steps and irradiation times. My opinion is that all these issues should be thoroughly explored to see in which extent they may significantly (or not) impact the dose estimates.

We do not think that the laboratory working time for SAR is shorter than MAAD, because we do X-ray irradiation, preheats and ESR measurements sequentially in the laboratory. Since SAR estimates De by an interpolation of a dose response curve, it does not need many number of dose points. Only the significant issue, which can affect the accuracy of the dose measurement is the potential sensitivity change by the annealing step, which was tested by the dose recovery test.

3. Residual dose

This might go beyond the scope of the present paper, and would perhaps deserve a proper consensus within the ESR dating community, but I am personally struggling with the use of the term 'residual dose', for two reasons. First, I am not sure whether the term is actually correct: technically, it should rather be a 'residual signal', which can then be converted into a dose value. Second, I find the term 'residual dose' misleading, as it may correspond to either (i) the unbleachable component of the Al signal, or (ii) the remaining component of the Al and Ti signals due to incomplete reset prior to deposition, (iii) or sometimes even both.

Moreover, the comparison of residual dose values from the various signals may be biased by the unbleachable component of the Al signal (see specific comments #8 & #9). This one should be subtracted to the so-called 'residual dose' in order to make it directly comparable with those from the Ti centres. Actually, I am not sure why the magnitude of this residual unbleachable component of the Al signal has not been evaluated in the present study using a sumlight simulator, so that it can be subtracted from the signal of the Natural and Gamma-irradiated aliquots. And I wonder in which extent this unbleachable component of the Al signal may impact the dose recovery experiment (and the poor ratio obtained), since this component cannot be optically reset (but thermally yes).

We have clarified the definition of the residual dose for the Al centre in this paper always includes the unbleachable part of the centre in the introduction. The reason we did not evaluate the unbleachable part of the Al centre was that there is no need if there are appropriate modern analogue samples are analysed and the residual dose from such samples were subtracted.

**Specific comments**

1. Abstract:

- l.4: does the value of 1350 Gy obtained for the Al centre include the unbleacheable component of the signal? if so, this should be mentioned.

Yes, the unbleachable signal part is included. This information has been added to the abstract.

- l.8 to 10: I would suggest to add a sentence indicating that the dose recovery ratios obtained for Al and Ti-H signals suggest dose estimates may be significantly overestimated and underestimated, respectively.

This information has been added to the abstract.

- l.9 to 11: Again, I would suggest some moderation in the final sentence: the results obtained by the authors indicate that the fluvial sediment samples **that have been investigated in this work** carry a significant residual dose, but this conclusion should not be extrapolated to all samples in any context. And especially for the Al and Ti-H signals given their dose recovery ratios.

The reviewer is right. This fact has been clarified in the text now.

2. l.20-22: there is also significant variability among the samples for the bleaching kinetics of the Ti-H and Ti-Li signals (see Tissoux et al., 2007; Duval et al., 2017). This may be mentioned.

*This information has been added to the text.*

3. l.22-23: The two parts of the sentence are not incompatible. The signal may be fully bleachable and incompletely bleached due to insufficient exposure to sunlight. Please rephrase without the 'although'.

*The sentence has been split and 'although' has been removed.*

4. l.24: studied or reported? All the publications from the 'French' group systematically provide the so-called 'bleaching coefficient' for the Al centre, which is the relative intensity of the unbleachable component (e.g., Voinchet et al., 2020; Duval et al., 2017, 2020a).

*We have mentioned Voinchet et al. 2015 about the bleaching index and residual doses after subtracting the unbleachbale residual of the Al center now in the MS (section 1).*

5. l.24: use 'residual signal' instead of 'residual dose'

*The aim is to quantify the residual and to make it comparable to other studies (Tsukamoto et al., 2017 and Timar-Garbor et al., 2020). Therefore, dose is the chosen unit.*

6. l.24, 'So far very few studies have reported residual doses of the quartz ESR signals from sediments': just to be clear, you mean residual component due to the incomplete reset of the ESR signal prior to sediment deposition?

*Yes, this is what we are referring to. The sentence has been rephrased to, "So far very few studies have reported residual doses of the quartz ESR signals from young or modern analogue samples, which could be directly comparable with the quartz OSL De values."*

l.26: again, technically, these are not residual doses' but rather 'residual signal that correspond to dose values of XXX'.

*The aim is to quantify the ESR residue, to make it comparable, and not just to report a dimensionless unit of intensity. Regardless, since there is no clear definition in the scientific community regarding the term "residual dose" it should be specified in the text, as the reviewer stated above, which of the three possibilities is meant. Here, the definition is given in the context of the paper itself and the papers, which we refer to.*

7. l.27-29: Question: does the so-called 'residual dose' reported by Tsukamoto et al. (2017) for the Al centre includes both unbleachable and unbleached (due to incomplete reset) components ? This should be clarified, otherwise these large numbers provided for the Al centre may be misleading.

*The residual doses of the Al centre reported by Tsukamoto et al. (2017) include the unbleachable signal part. This information has been added to the text now to clarify.*

8. l.28-32: I believe the same comment applies to the study by Timar-Gabor et al (2020): in order to be discuss values that are directly comparable for the Al and Ti centres, the author should report the so-called residual dose for the Al signal without the unbleachable component. Otherwise, this may be again misleading: the huge apparent residual dose for

the Al centre is actually made of 2 components, with one of them that cannot be optically reset. The standard procedure in the dose evaluation associated with the Al centre does include the subtraction of the unbleachable component of the signal (e.g. Voinchet et al., 2020; Duval et al., 2017, 2020a)

The determination of the unbleachable signal part requires extra ESR spectrometer time and extensive solar simulator time. Therefore, it may not always be possible to determine the unbleachable signal part of the Al centre. If it is noted that the information given is the total dose/total signal, this should be regarded as sufficient.

9. l.34: again, what does residual dose mean here? The magnitude of the unbleachable component is systematically reported in all the studies mentioned (cf., 'bleaching coefficient' in the tables).

This has already been answered to in 5.-7.

10. l.35: Toyoda and al do not report ESR any age estimates in that paper.

This has been corrected to ESR intensities.

11. l.45: why for the first time? The authors report many previous studies with a similar approach (Beerten et al. 2006; Tsukamoto et al., 2017; Timar-Gabor et al., 2020

We report ESR residual doses for the first time for a fluvial sedimentary environment. Tsukamoto et al., 2017 and Timar-Gabor et al., 2020 report residual doses for aeolian environments. Beerten et al., 2006 QG in fact state for sample MS9 with total Ti-Li De of 53+-5 Gy "The existence of a significant nonzero Ti–Li equivalent dose for the youngest sample (MS9) strongly indicates the presence of an unbleached or unbleachable residual dose in this sample." But again, aeolian deposits.

12. l.48, section 2. Samples:

- Just checking, the samples analysed in the present study are the same that have been analysed earlier by Lauer et al (2011): i.e., these are the exact same prepared quartz ? or do they come from a another sample preparation? This should be clarified in the MS.

Yes, the exact same samples as used by Lauer et al. (2011) have been used. No additional preparation has been done. This has been clarified in the text now (section "Samples" of the MS).

- l.48: The authors should provide a basic description of the sedimentary environment (facies, grain size) for each sample: these parameters are known to impact the reset of the ESR signals (e.g., Voinchet et al., 2015)

The information of the grain size has been added to the MS (see below). We added a table with a brief description of the samples (Table 1) and a cross reference to the Lauer et al. 2011 paper to section "Samples" of the MS.

- l.48: gain size of the samples dated ? (this can be added to Table 2)

The grain size for all the samples were 100-250 microns (same as for the luminescence dating carried out by Lauer et al., 2011). This information has been added to section "Samples" of the MS.

- l.50-55: Unfortunately I could not access the paper by Lauer et al. (2011). Would be good if the authors could provide some additional info about how the OSL De values were calculated (single grain? Single aliquot? How many aliquots/grains measured to get the mean De values? etc.)

The SAR protocol has been used by Lauer et al. (2011) to obtain $D_e$. This information has been added to the section "samples" of the MS. Lauer et al. (2011) do not provide any information of how many aliquots were used to get the mean $D_e$ values form OSL measurements, but from their diagrams, it must have been ~50 small aliquots per sample were measured.

14. l.59, Section 3. ESR measurements:

- l.65: 'modulation amplification' or 'modulation amplitude'?

The text has been corrected to "modulation amplitude".

- l.66: remove 'of the spectra'

'of the spectra' has been removed in this sentence.

- l.68: what does 'homogenous value' mean? What about rephrasing as 'to take into account the angular dependence of the signal' instead ?

Yes, that is what it was supposed to say. For clarification, the text has been changed to the reviewer's suggestion.

- l.69: a microwave power of 10mW for the Ti signals seems too much. Have the authors performed a microwave saturation curve ? Our data show that the Ti signals saturate at such power, which is why we usually measure using 5 mW.

Previously, we measured the microwave power dependency on a sample of Chinese loess and did not observe any saturation up to 20 mW for the Ti centre.

- l.74: that is correct, and therefore the peak @ g=1.915 does not result from the single contribution of the Ti-Li signal, but from a mixture of both centres. This should be reported in this way throughout the manuscript.

Other authors also used the term "Ti-Li" (e.g. Tissoux et al. 2008). In the MS, we clarify that "Ti-Li" this signal is also affected by Ti-H. We added a reference to the Duval and Guilarte 2015 paper labelling of the Ti centres.

- l.79: What is this reference 'Tsukamoto, 2019, unpublished'? Is this a technical report? Not sure this should be mentioned here as it cannot be accessed.

A paper containing this information has been already submitted to RM and is in review stage at the moment. Therefore we changed the reference to "Tsukamoto et al, submitted".

18. l. 99, Subsection Equivalent doses, residual doses and ESR ages:

- l.109: I am not sure in which extent the OSL dose rate can be directly used for ESR. Maybe they should be adjusted. There are some specificities, like the alpha efficiency of 0.07 (Bartz

et al., 2019b), that should probably be taken into account for a small (but non-negligible) internal dose rate and external alpha dose rate components.

Dose rate is the dose absorbed by the host material. We think it is logical to use the same dose rate as the quartz OSL.

- l.111: 'These residual ages show how significant the effect of the residual dose **may be** in ESR dating of fluvial sediments.' This is valid for the present set of samples. Not all samples behave the same.

The text has been changed to the reviewer's suggestion.

19. Subsection Dose recovery test, l.121: in which extent the unbleachable component of the Al signal may impact the dose recovery ratio ?

This is a good question, but we do not know yet.

20. l.127:, 'The results clearly show that the ESR De for all samples are significantly larger than the OSL De of Lauer et al. (2011) and therefore residual subtraction is highly recommended'. 2 comments here: (1) if this residual component resulting from incomplete bleaching of the signal is not subtracted, then the ESR age results should be regarded as maximum possible burial ages; (2) residual subtraction would require the use of modern-analogue samples, but this should not be regarded as the perfect solution: it is based on the assumption that modern-analogues and Pleistocene samples that are being dated have experienced similar transport and bleaching conditions, which can not always be verified. Therefore, subtracting the 'residual' signal derived from modern-analogue samples may introduce another source of uncertainty in the equation, and does not ensure that the ESR signal of the dated sample has actually been fully reset.

We also do not think that the residual subtraction using modern analogue would be a perfect solution. However, we obtained relatively uniform residual doses at least from from the Ti centre (Ti-Li and mix) and therefore this value can be used for instance, the higher terrace sediments. This is at least a better approach than not subtracting.

21. Figure 1: I think there is an issue with the peaks reported for the Ti-H signal. There should be doublets for each g value, and there is actually an overlap on the Ti-Li peak @1.913 (e.g. Toyoda et al., 2000).

The indictor for peak $g_3$ for the Ti-H signal has been corrected and the indicators for the doublets have been added to Figure 1.

22. Figures 3 and 5: does the Al residual De values include the unbleachable component ? I believe it should be subtracted, in order to make the all data really comparable.

Yes, the $D_e$ values for the Al centre in Figure 3 and 5 include the unbleachable component. We did not estimate the size of the residual signal by a bleaching test, because if we subtract a residual dose, preferably obtained from a same set of sedimentary sequence, there is no need to determine the size of bleachable/unbleachable components of the Al centre. We consider it is a huge advantage of this approach, considering the fact that we needs to change a lamp of the solar simulator every time we do a bleaching experiment (for 1000 hours).

Table 1, footnote: *including unbleachable signal component

The footnote in Table 1 (and 2) has been changed to the reviewer's suggestion.

**References**

Beerten, K., Koen Verbeeck, Eric Laloy, Veerle Vanacker, Dimitri Vandenberghe, Marcus Christl, Johan De Grave, Laurent Wouters (2020) Electron spin resonance (ESR), optically stimulated luminescence (OSL) and cosmogenic radionuclide (CRN) dating of quartz from a Plio-Pleistocene sandy formation in the Campine area, NE Belgium. Quaternary International.

Bartz, M., Arnold, L.J., Demuro, M., Duval, M., King, G.E., Álvarez Posada, C., Parés, J.M., Rixhon, G. and H. Brückner (2019a). Single-grain TT-OSL dating results confirm an Early Pleistocene age for the lower Moulouya River deposits (NE Morocco). Quaternary Geochronology 49, pp. 138-145.

Bartz M., Arnold L.J., Spooner N., Demuro, M., Campaña Lozano I., Rixhon G., Brückner H., Duval M. (2019b). First experimental evaluation of the alpha efficiency in coarse-grained quartz for ESR dating purpose: implications for dose rate evaluation. Scientific reports 9:19769, pp. 1-10. https://doi.org/10.1038/s41598-019-54688-9.

Duval, M. and Guilarte, V. (2015). ESR dosimetry of optically bleached quartz grains extracted from Plio-Quaternary sediment: Evaluating some key aspects of the ESR signals associated to the Ti-centers. Radiation Measurements 78(0): 28-41.

Duval, M., Arnold, L.J., Guilarte, V., Demuro, M, Santonja, M., Pérez-González, A. (2017). Electron Spin Resonance dating of optically bleached quartz grains from the Middle Palaeolithic site of Cuesta de la Bajada (Spain) using the multiple centres approach. Quaternary Geochronology 37, pp. 82-96.

Duval M., Voinchet P., Arnold L.J., Parés J.M., Minnella W., Guilarte V., Demuro M., Falguères C., Bahain J.-J. & Despriée J. (2020a). A multi-technique dating study of two Lower Palaeolithic sites from the Cher Valley (Middle Loire Catchment, France): Lunery-la Terre-des-Sablons and Brinay-la Noira. Quaternary International 556, pp. 71-87. DOI: 10.1016/j.quaint.2020.05.033.

Duval, M., Arnold, L.J., Rixhon G. (2020b). Electron Spin Resonance Dating in Quaternary studies: evolution, recent advances and applications. Quaternary International 556, pp. 1-10. Guest Editorial. DOI: 10.1016/j.quaint.2020.07.044.

Rink, W.J., Bartoll, J., Schwarcz, H.P., Shane, P. and Bar-Yosef, O. (2007). "Testing the reliability of ESR dating of optically exposed buried quartz sediments." Radiation Measurements 42(10): 1618-1626.

Timar-Gabor, A., Chrúscinska, A., Benzid, K., Fitzsimmons, K.E., Begy, R. and Bailey, M. (2020). Bleaching studies on Al-hole ([AlO4/h]0) electron spin resonance (ESR) signal in sedimentary quartz.Radiation Measurements 130: 106221.

Tissoux, H., Falguères, C., Voinchet, P., Toyoda, S., Bahain, J.J. and Despriée, J. (2007). Potential use of Ti-center in ESR dating of fluvial sediment. Quaternary Geochronology 2(1–4): 367-372.

Toyoda, S., Voinchet, P., Falguères, C., Dolo, J.M. and Laurent, M. (2000). Bleaching of ESR signals by the sunlight: a laboratory experiment for establishing the ESR dating of sediments. Applied Radiation and Isotopes 52(5): 1357-1362.

Voinchet, P., Toyoda, S., Falguères, C., Hernandez, M., Tissoux, H., Moreno, D. and Bahain, J.J. (2015). "Evaluation of ESR residual dose in quartz modern samples, an investigation on environmental dependence." Quaternary Geochronology 30, Part B: 506-512.

Voinchet, P., Pereira, A., Nomade, S., Falguères, C., Biddittu, I., Piperno, M., Moncel, M.H. and Bahain, J.J. (2020). "ESR dating applied to optically bleached quartz - A comparison with 40Ar/39Ar chronologies on Italian Middle Pleistocene sequences." Quaternary International.

Mathieu Duval, Burgos, 14/05/2021.

PS: not sure why, but the numbering of the sections and bullet points changes from the Edit to the Preview versions. Hope this can be fixed.

We also do not know why, we hope this will not occur again.

---

## Author Response (AR2)

**Suggestions for revision or reasons for rejection (will be published if the paper is accepted for final publication)**

As mentioned in my first review the study "Investigation of quartz ESR residual signals in the last glacial and early Holocene fluvial deposits from the Lower Rhine" is interesting and worth being published. I congratulate the authors for their work. Furthermore, the manuscript was significantly improved following the reviewing process.

We thank the reviewer for the kind words on the improvement of the manuscript.

I have only one comment left. I am uneasy with the term "sensitivity change". I find the term a "black box". While in luminescence we have at least models of what might be found in that back box, in ESR, in my personal view, there are no models put forward yet. As such, I find it luminescence jargon transposed without a clear basis yet. By this comment it is not my intention to impose to the authors not to use the term, but as an intro to the dose recovery test interpretation that I am still not fully satisfied with. In the introduction the authors state that "Previously, ESR dose recovery tests have been conducted by Beerten et al. (2008) on quartz derived from dune sands and Asagoe et al. (2011), who used quartz from tephra samples. Unfortunately, both studies use an intensive thermal treatment (annealing) of the sample to erase the natural signal before artificial irradiation, which reduces the significance of the test." However, here a preheat is applied as well. Moreover, an unknown value (the De) is fed into the protocol as a known one, as explained in my previous comment. While in principle I agree with the preheat, I am still not convinced that the way this dose recovery test was conducted is the best possible one. Maybe it would have been better to subject the sample to extensive bleaching experiments until (hopefully) a constant residual would be reached, and then add the 1000 Gy dose. I understand this is extremely time consuming and I do not impose it now either. However, I think a caveat in this regard should be added in the text. In my view, while dose recovery tests should be incorporated into ESR dating, there is still work to be done before an exact protocol is established.

To avoid any misunderstandings, we have deleted most of the phrase "sensitivity change" from the manuscript and explained in other words.

I do not fully grasp what is meant by "However, the result of the dose recovery test suggests that the ratio of bleachable/unbleachable components should be compared before and after the annealing step, in order to understand the problem of the dose recovery test." in the discussion. It might be better to move this information to the dose recovery section and expand it for clarity.

We have added a more detailed explanation in the discussion section of the paper.

Other small corrections:
-paragraph at line 75 subscripts and superscripts are missing: "$[AlO_4]^0$ $[TiO_4/M^+]^0$"

The missing subscripts and superscripts have been added.

- paragraph at line 135 "thermal treatment included in the SAR protocol may change sensitivity of the ESR centres per unit dose" delete per unit dose as sensitivity itself is signal per unit dose.

We rephrased the sentence without using "sentitivity".

**Suggestions for revision or reasons for rejection (will be published if the paper is accepted for final publication)**

The manuscript focuses on residual ESR signals in quartz (Al and Ti centres) of fluvial deposits from the lower Rhine. In this study, the authors compare newly obtained ESR results with OSL ages (Lauer et al., 2011). In general, the manuscript is well written and has improved based on the reviewer comments. I think this manuscript is definitely worth to publish as it gives valuable information in ESR signals in a fluvial environment, which is important for other ESR dating studies. However, I think the authors can extend the discussion and focus a bit more on the ESR signal properties rather than only presenting residual doses, which might not be fully reliable (poor DRT, sensitivity changes). My main concern focuses on the poor dose recovery test.

We thank the reviewer for the very insightful comments to our manuscript.

1) The dose recovery test is an issue in my opinion. How can we trust a measurement protocol that is proven inaccurate based on the poor DRT?
a) What are the fitting uncertainties of the different centres?

In terms of the fitting uncertainties for the DRT, the $R^2$ for the Al centre is $0.993 \pm 0.002$, for the Ti-Li centre is $0.995 \pm 0.001$, for the Ti-mix centre is $0.999 \pm 0.001$ and $0.994 \pm 0.006$ for Ti-H centre.

I wonder whether the poor DRT results from the Ti-H centre come from a poor measurement (e.g. low ESR intensity). I appreciate that the authors provided additional information on the measurement uncertainties. I think this is an important point. The Ti-H centre shows an uncertainty of up to 16% and just between the three rotations. It was noted that the ESR intensity of the Ti-H centre was "relatively small".

I think, in terms of Ti-H and the large uncertainty is it a mixture of the weak natural signal, which results in a low S/N ratio, an angular dependency of the signal itself, and for sure, to a certain portion, human error.

The authors should discuss the reliability of the different ESR signals regarding signal properties and experimental errors. At the moment, the manuscript does not provide any information on the reliability of the signals, except of poor DRT results (Al and Ti-H). Please provide more information on ESR intensities, fitting uncertainties, etc. in the manuscript or SI.

For sample RB-II which was used for DRT, the Al centre signal intensity (a.u.) is ~8 times bigger than Ti-Li, ~7 times bigger than Ti-mix and ~22 times the size of Ti-H.
In terms of the fitting uncertainties for the DRC's (DRT), the $R^2$ for the Al centre is $0.993 \pm 0.002$, for the Ti-Li centre is $0.995 \pm 0.001$, for the Ti-mix centre is $0.999 \pm 0.001$ and $0.994 \pm 0.006$ for Ti-H centre. The relative standard deviation (1-sigma, %) of the three measurements for sample RB-II natural signal were for Al 0.55, Al after subtraction of peroxy 0.86, Ti-Li 2.27, Ti-mix 2.9 and Ti-H 16.1.

b) What is the saturation level of the Ti-H centre? The added ca. 1000 Gy on top of the natural (residual) signal is in my opinion way too much for the Ti-H centre, which saturates usually much earlier than the Al and Ti-Li centres and often <1000 Gy. Can the DRT work if the Ti-H is saturated? It would be worth showing the D0 values of the signals.

In terms of the sample used for the DRT, the mean $D_0$ for Ti-H is 668 Gy. So it is close to 2x $D_0$ after adding 963 Gy, but still below 2x$D_0$.

2) How do sensitivity changes affect the results? The authors mentioned "Only the significant issue, which can affect the accuracy of the dose measurement is the potential sensitivity change by the annealing step, which was tested by the dose recovery test." and the "reported residual doses may be overestimated [Al centre]" due to sensitivity changes. In line 105, an annealing treatment of 420 °C for 2 min was chosen based on the results of Tsukamoto et al. (2015), which did not show sensitivity changes. In contrast, sensitivity changes were observed in the current study. To which extent are the Al dose results overestimated due to sensitivity change (75%?)? It should be discussed better in regard to the final residual dose results as a recommendation is given in the conclusion to subtract such high doses.

If the SARA slope is 1.75, yes you can correct the $D_e$ by dividing the $D_e$ with the slope; this is the definition of SARA. But since we only use one added dose point in our dose recovery experiment, we do not want to do this. At the moment, even with the Ti centre (better bleachable signal than the Al centre) we observed significant residual dose – all we can say is using the Al centre to date fluvial sediments is even more difficult.

3) Inter-aliquot scatter is one point, but technical uncertainties (e.g., differences in spectrometer sensitivity, HS cavity) might lead to variations in De repeatability. This has often been observed in other ESR dating studies using even larger aliquot sizes (e.g. 200 mg). I know that ESR measurements are very time consuming, but I think it would be worth exploring the variability in De values of different aliquots for a single sample. I agree that in the present study the focus lies on the determination of the residual dose, and not on a precise burial date, but given the fact that the residual dose should be subtracted from the sample's De to receive a more precise age, I think it is important to evaluate also the residual dose in a more robust way. Maybe the authors could give a recommendation for ESR measurements in view of future single aliquot measurements. For example how many aliquots should be measured in order to provide a better statistical estimate of the De (otherwise it will be just one De value that might not be correct).

We do not really think it is possible to obtain a precise residual dose from modern analogue samples. Our result from different samples using one aliquot each from the same terrace is similar enough to give a ball park residual dose from the Rhine sediments. As we did not measure multiple single aliquots for the samples, it is not possible to make such recommendation in this paper.

4) Although the authors have included Table 1 showing a brief sample description, the manuscript still does not show any discussion on sediment process properties of the samples in relation to their bleaching kinetics. As sediment properties seem to be different (Table 1) and residual dose variations have been observed (e.g., Ti-Li ~400-900 Gy), it would be interesting if the authors could comment on this in the discussion, highlighting bleaching differences of different fluvial transport modes. It is important which kind of fluvial deposit was measured as the transport mode has a large impact on the bleaching of the signals. More information should be given in the discussion.

We have added a paragraph on the samples' deposition environments and their bleaching kinetics with respect to the ESR residual doses we measured to the discussion part of the MS. We did not find any dependency of the ESR residual with the deposition environments.

5) Line 148: I agree that the subtraction of a total residual might give us an idea about the bleached ESR signal. However, it remains difficult to find a representative modern analogue for Pleistocene samples, and if there is no appropriate sample available we still deal with two unknowns: the partially bleached component and the optically unbleachable component. So I think it is difficult to get a reliable residual dose that can be subtracted from older samples, and measuring an artificially bleached aliquot to estimate at least the unbleachable Al component is important. "residual subtraction is highly recommended" (line 148) should be rephrased, as this is only valid when a representative modern analogue sample is available.

The sentence has been rephrased according to the reviewer's suggestion: "The results clearly show that the ESR $D_e$ for all samples are significantly larger than the OSL $D_e$ of Lauer et al. (2011) and therefore residual subtraction is highly recommended if a representative modern analogue sample is available."

6) Lines 148-150: „the observed residual doses confirm the trend in the signal's bleaching behaviour". This is only true when the results of all centres are reliable, but as mentioned before the Al is probably overestimated and the Ti-H centre results are probably underestimated. Moreover, the total residual dose of the Al centre contains two components (optically unbleachable component and remnant dose due to partial optical bleaching), so the Al centre is only hardly comparable to the Ti centres as the latter does not contain an unbleachable signal component. Secondly, the Al centre is characterised by slower bleaching kinetics.

We have rephrased the sentence to "Furthermore, the observed residual doses follow the trend in the signal's bleaching behaviour as described by Toyoda et al. (2000): the Al centre shows the largest residual followed by the Ti-Li and Ti-H with the lowest. The size of the residual dose for the Ti-mix lies in between the Ti-Li and Ti-H."

The rough estimate of the size of the unbleachable Al component (line 160) is 90±210 Gy? How is this in agreement with estimations of 500-700 Gy by Tsukamoto et al. (2018) and Timar-Gabor et al. (2020) (line 161)? It remains rather difficult to estimate the unbleachable component without a proper bleaching test. Please clarify that in the discussion.

For clarification, the aforementioned sentence has been extended by "[…], although it is much smaller than the values reported by Tsukamoto et al. (2018) and Timar-Gabor et al. (2020) from aeolian sediments."

---

## Author Response (AR3)

Please, if possible, change the orientation of tables 2 and 3 to portrait. Furthermore, Supplement figure should be renamed according to the standards to Figure S1

Table 2 was set up in portrait orientation. Table 3 is too wide to fit in A4 portrait orientation. Therefore, I used the "sidewaystable" function provided by the Latex template. The numbering of the supplementary figure was changed to Figure S1. This was also changed in the text of the manuscript.